# High Resolution Episcopic Microscopy for Qualitative and Quantitative Data in Phenotyping Altered Embryos and Adult Mice Using the New “Histo3D” System

**DOI:** 10.3390/biomedicines9070767

**Published:** 2021-07-01

**Authors:** Olivia Wendling, Didier Hentsch, Hugues Jacobs, Nicolas Lemercier, Serge Taubert, Fabien Pertuy, Jean-Luc Vonesch, Tania Sorg, Michela Di Michele, Laurent Le Cam, Thomas Rosahl, Ester Carballo-Jane, Mindy Liu, James Mu, Manuel Mark, Yann Herault

**Affiliations:** 1CNRS, INSERM, CELPHEDIA, PHENOMIN-Institut Clinique de la Souris (ICS), Université de Strasbourg, 1 Rue Laurent Fries, 67404 Illkirch Graffenstaden, France; olivia@igbmc.fr (O.W.); hugues@igbmc.fr (H.J.); fabien.pertuy@alumni.unistra.fr (F.P.); tsorg@igbmc.fr (T.S.); marek@igbmc.fr (M.M.); 2CNRS, INSERM, Institut de Génétique et de Biologie Moléculaire et Cellulaire (IGBMC), Université de Strasbourg, 1 Rue Laurent Fries, 67404 Illkirch Graffenstaden, France; dihe@igbmc.fr (D.H.); taubert@igbmc.fr (S.T.); jlv@igbmc.fr (J.-L.V.); 3Imagxcell, 2 Allée des Feuillantines, 94800 Villejuif, France; nicolas.lemercier@imagxcell.com; 4Institut de Recherche en Cancérologie de Montpellier (IRCM), INSERM U1194, Université Montpellier, 34298 Montpellier, France; michela.dimichele@inserm.fr (M.D.M.); laurent.lecam@inserm.fr (L.L.C.); 5Institut Régional du Cancer de Montpellier (ICM), Université Montpellier, 34298 Montpellier, France; 6Merck & Co. Inc., Kenilworth, NJ 07033, USA; thomas_rosahl@merck.com (T.R.); ester_carballojane@merck.com (E.C.-J.); dongming.liu@merck.com (M.L.); yingjun_mu@merck.com (J.M.); 7Service de Biologie de la Reproduction, Hôpitaux Universitaires de Strasbourg (HUS), CEDEX, 67091 Strasbourg, France

**Keywords:** 3D imaging, high-resolution episcopic microscopy, phenotyping, microanatomy, quantification

## Abstract

3D imaging in animal models, during development or in adults, facilitates the identification of structural morphological changes that cannot be achieved with traditional 2D histological staining. Through the reconstruction of whole embryos or a region-of-interest, specific changes are better delimited and can be easily quantified. We focused here on high-resolution episcopic microscopy (HREM), and its potential for visualizing and quantifying the organ systems of normal and genetically altered embryos and adult organisms. Although the technique is based on episcopic images, these are of high resolution and are close to histological quality. The images reflect the tissue structure and densities revealed by histology, albeit in a grayscale color map. HREM technology permits researchers to take advantage of serial 2D aligned stacks of images to perform 3D reconstructions. Three-dimensional visualization allows for an appreciation of topology and morphology that is difficult to achieve with classical histological studies. The nature of the data lends itself to novel forms of computational analysis that permit the accurate quantitation and comparison of individual embryos in a manner that is impossible with histology. Here, we have developed a new HREM prototype consisting of the assembly of a Leica Biosystems Nanocut rotary microtome with optics and a camera. We describe some examples of applications in the prenatal and adult lifestage of the mouse to show the added value of HREM for phenotyping experimental cohorts to compare and quantify structure volumes. At prenatal stages, segmentations and 3D reconstructions allowed the quantification of neural tissue and ventricular system volumes of normal brains at E14.5 and E16.5 stages. 3D representations of normal cranial and peripheric nerves at E15.5 and of the normal urogenital system from stages E11.5 to E14.5 were also performed. We also present a methodology to quantify the volume of the atherosclerotic plaques of *ApoE^tm1Unc/tm1Unc^* mutant mice and illustrate a 3D reconstruction of knee ligaments in adult mice.

## 1. Introduction

Genetically engineered mice (GEM) play a central role in the functional annotation of the human genome. They are used extensively to study the role of genetics in normal development, as well as the molecular pathogenesis of disease and ultimately in the evaluation of treatment effects for disease models. Many GEMs exhibit gestational lethality [1,2]. The foundation of phenotypic analysis in developing mice is qualitative assessment for structural defects and functional deficiencies. However, quantitative methods are often needed to measure the dimensions, volume, or cell numbers in particular structures. Morphometric data illustrate the severity of the observed abnormalities (mild, minimal, moderate, and marked) but also demonstrate the presence of subtle defects that are not detectable by classical histology.

Morphologic changes in mouse mutants are usually examined by means of classical histological methods. The generation of sections in standard orientations allows for qualitative and descriptive analysis. However, additional techniques for understanding more complex morphological processes and allowing high-resolution imaging of an entire organ or embryo may be more informative and complementary. Histological section-based methods can provide three-dimensional (3D) information on tissue architecture, but these techniques are labor-intensive and time consuming and the quality of the 3D models is low due to mechanical distortions and difficulties in the alignment of the resulting images [3,4,5]. 3D imaging modalities such as micromagnetic resonance imaging (µMRI) [6,7] and microcomputed tomography (µCT) [8,9,10,11,12] are promising techniques but have limited resolutions. Additionally, technological advances in microscopy have facilitated a dramatic increase in the imaging of high-resolution, intact whole organs using serial two- photon tomography (STPT) [13] or tissue clearing techniques and lightsheet microscopy [14,15,16] or optical projection tomography (OPT) [17,18,19].

High-resolution episcopic microscopy (HREM) combined with 3D analytical technique is an interesting alternative. HREM offers volumetric data sets composed of consecutive digital images obtained from the block surface of methacrylate resin-embedded samples that are serially sectioned [20,21,22]. The resin is supplemented with eosin and acridine orange to provide contrast to highlight the morphology of the tissue and the digital data that are produced match the quality of a digital image captured from a histological section. Since the images are captured from the surface of the block of resin (rather than from glass-mounted sections), the set is inherently complete, aligned and unfolded. HREM volume data can be immediately analyzed with orthogonal and oblique virtual resectioning tools and appropriate software. This approach generates very high-resolution reconstructions of large specimens with excellent shape preservation. HREM has proven to be an excellent tool in a broad variety of research fields [23,24]. In addition to its use for screening the phenotype of prenatally lethal E14.5/E15.5 mouse embryos produced in the framework of the International Mouse Phenotyping Consortium (IMPC) program [25,26,27], HREM has also been employed for visualizing tissue samples of adult biomedical model organisms and human skin biopsies [28,29].

We have developed a new apparatus called “Histo3D” with the LEICA HistoCore Nanocut automated rotary microtome for HREM experiments. Here, we describe some examples of applications in the prenatal and adult life stage of the mouse to show how HREM technology is a helpful method in the study of microanatomy and is of great value for phenotyping experimental cohorts in order to quantify structure volumes over the time course of development. We present a methodology to quantify the neuroepithelium and ventricular system volumes in mouse fetuses at stages E14.5 and E16.5, and to describe morphological changes of cranial nerves and ganglia at E10.5, E11.5 and E15.5, and of the urogenital system in female mouse embryos occurring between E11.5 to E14.5. We also present a method to quantify atherosclerotic plaques on hypercholesterolemic mice and describe the insertion of ligaments in the 3D reconstructions of an adult knee joint.

## 2. Materials and Methods

### 2.1. Sample Preparation and HREM Data Generation

The samples were fixed in Bouin’s fixative for 24 h minimum, then washed and stored in 70% EtOH. Fixed samples were gradually dehydrated in an increasing series of ethanol concentrations (90%, 95%, 100%, 3 h each) and were embedded in a methacrylate resin (JB-4 kit, Polysciences, Warrington, PA, USA) containing eosin and acridin orange as contrasts agents, as previously described; [23,30,31] https://dmdd.org.uk/hrem/, accessed on 28 June 2021). The typical sizes of commercial polyethylene embedding molds are 6 × 8 × 15 mm^3^ or 13 × 19 × 15 mm^3^ (HistoMold, Biosystems). A 1–2-mm cushion of polymerized resin was used to ensure that samples remained entirely embedded within the resin without coming in contact with the block surface. Blocks were left to polymerize overnight at 4 °C; they were taken out of the molds before being baked at 90 °C overnight to ensure a uniform, hard texture prior to sectioning. Samples were subsequently cooled and stored at room temperature for no more than one week before imaging. Before sectioning, they were stuck on block holders with cyanoacrylate glue. It is noteworthy that the size and type of samples can be critical for correct embedding with JB4 resin. The size of the sample should not exceed 15 × 15 mm^2^ and sometimes even if they are smaller, some samples can present embedding artifacts (bubbles) due to abnormal polymerization.

The resin blocks were mounted on the Histo3D system and initially trimmed without imaging to ensure that the block surface was exactly perpendicular to the optical axis (see Figure 1 for technical details). Once a perfectly flat surface was obtained, the block was removed from the microtome to localize the sample with a lamp, and we evaluated its size and location with a marker. This step is required in order to establish the field of view since it has been calibrated with a micrometer rule for each specific position of the manual zoom (see Table 1) to establish the numerical resolution (in voxel size). The focusing of the surface plane was obtained by drawing lines on the surface with a worn thick marker.

### 2.2. Image Analysis and Quantification

2-D raw data were imported into Avizo visualization software (Version 9.4.0, Thermofisher Scientific, Bordeaux, France) for volumetric visualization, segmentation, and quantitative analysis. Before importation, the images were cropped, eventually scaled and contrast optimized using Image J. No realignment of images was necessary due to the precise positioning of the block after each section. Each dataset was subsampled in Avizo to obtain cubic voxels, allowing orthogonal and oblique views of the tissue sample. Automatic segmentation was performed using the Watershed algorithm. Manual segmentations were performed using the brush, lasso, or magic wand (region growing) tools in Avizo, according to the structure to be outlined. After segmentation, the Avizo software generated a mask that allowed us to obtain the quantitative data.

#### 2.2.1. Embryo Data Generation

All embryos were oriented sagittally in the mold. Embryos from E10.5 and E11.5 stages were sectioned to generate 5-µm-thick sections. The data consisted of around 300 and 400 images, respectively, with a voxel size of 3.2 × 3.2 × 5 µm^3^. Embryos from E12.5 to E15.5 were sectioned at 7 µm. The number of images was 600, 700, 800 and 950, respectively, with voxel sizes of 5.2 × 5.2 × 7 µm^3^ for E12.5 embryos, 6.5 × 6.5 × 7 µm^3^ for E13.5 embryos and 8 × 8 × 7 µm^3^ for E14.5 and E15.5 embryos. At the E16.5 stage, only the heads of the embryos were processed. They were sectioned at 7 µm and had a resolution of 6.5 × 6.5 × 7 µm^3^ and generated around 800 images. The duration of sectioning ranged from 1 h for E10.5 embryos (4 × 3 × 2 mm^3^ in volume) to 3.5 h for E15.5 embryos (14 × 8 × 7 mm^3^). Automated segmentation using the Watershed tool was performed for the whole embryos or the whole brains. The nerves, ganglia and organs of the urogenital systems were segmented manually. The numerical data shown for the volume quantification of the neuroepithelium and ventricular system are represented in pixel size.

#### 2.2.2. Aorta Data Generation

During dissection, whole aortas were put on filter paper with an S shape to minimize their length. They were fixed with the paper to ensure flat embedding in the bottom of the mold. The total volume was around 14 × 1.5 × 1.5 mm^3^. The resulting HREM data had a voxel size of 8 × 8 × 7 µm^3^ and consisted of 200 aligned images obtained in one hour. Automatic segmentation was performed for the whole aortas and the atherosclerotic plaques were segmented manually. The numerical data shown for the quantification of aorta volumes are represented in pixel size.

#### 2.2.3. Knee Joint Data Generation

Knee joints were sectioned at 4 µm. The resulting HREM data had a voxel size of 6.5 × 6.5 × 4 µm^3^ and consisted of 1000 aligned single images. Automatic segmentation was performed for the bones and the ligaments were segmented manually.

## 3. Results

The assembly of the microtome-integrated Histo3D optical system is described in Figure 1. Briefly, the Histo3D consists of a system in which a zoom mounted with a camera is assembled on the Leica Biosystem Nanocut rotary microtome. The Histo3D pixel size and field of view for specific zoom positions are listed in Table 1. The novelty of the Nanocut microtome resides in its 3D optional mode and its optimization thanks to our collaboration with Leica. Compared to other rotary microtomes, it was modified to allow the block holder to reproducibly come to rest at an “imaging position” after each cut, permitting perfect image alignments. The stack of 2D images that are generated will directly permit the 3D rendering of the samples (here, a whole mouse embryo at E15.5) (Figure 1c). The contrast of the images was considerably improved by the presence of a custom Ploem fluorescence illuminator and the system is compact and easily upgradable.

### 3.1. Qualitative and Quantitative Morphological Assessment in Mouse Developmental Pathology Studies

#### 3.1.1. Neuroepithelium and Ventricular System Volume Quantification at Two Developmental Stages E14.5 and E16.5 in the Mouse

During brain development, both the neuroepithelium and embryonic cerebrospinal fluid (CSF) work in an integrated way to promote embryonic brain growth, morphogenesis and histogenesis [32,33]. Separating neuroepithelium and ventricular system quantitative analyses can point to subtle neurogenesis defects, because a smaller volume of the neuroepithelium and/or ventricles can predict abnormal neurogenesis. Several imaging techniques have been used to quantify the ventricular system during mouse development including ultrasound [34,35] and MRI [36]. Defects in embryonic neurogenesis in mouse models of Down syndrome have been linked to postnatal enlargement of the cerebral ventricles [37], and changes in ventricle volume have been observed in children with neurologic disorders such as Tourette syndrome [38].

We used Histo3D to evaluate the volumes of the neuroepithelium and the cerebral ventricles of genetically engineered mutant cohorts and their wild-type littermates at two embryonic stages: E14.5 and E16.5 (Le Cam, 2021 unpublished results). Here, we focused on the methodology and results obtained in the wild-type embryos. The neuroepithelium was segmented into the forebrain (itself subdivided in two parts: the telencephalon and diencephalon + hypothalamus), the midbrain and the hindbrain (see Figure 2a,b, Appendix A). These subdivisions were made according to the Allen brain atlas https://developingmouse.brain-map.org/static/atlas, accessed on 28 June 2020. The ventricular system was segmented (Figure 2a,c, Appendix A) into the lateral ventricles, third ventricle, mesencephalic vesicle (future cerebral aqueduct) and fourth ventricle. After manual segmentation, the Avizo software generated a mask that allowed the quantitative analysis of the volumes of the different structures.

Quantifications were performed on three age-matched wild-type embryos, chosen according to their body weight. The neuroepithelium and ventricular volumes between individuals followed the same pattern when expressed as absolute values or ratios relative to embryo (E14.5) or head (E16.5) volumes (Figure 2d,e and data not shown), showing that our quantification method is robust and that cohorts of three individuals of similar weight were sufficient to obtain statistically significant results illustrating structural abnormalities of the brain. We took advantage of the segmentations to compare the evolution of the different structures between E14.5 and E16.5 (Figure 2f,g). Thus, we could note that the neuroepithelium volume in E16.5 embryos was three times bigger than that of E14.5 embryos, whereas the ventricle volumes were relatively stable between these two stages. These analyses also highlighted that the telencephalon neuroepithelium increases by 3.3 times, whereas the diencephalon/hypothalamus only increased by 2.3 times between E14.5 and E16.5. These data illustrate the benefit of quantifying different parts of the brain rather than the whole brain in a phenotyping pipeline.

#### 3.1.2. 3D Reconstructions of Cranial Nerves and Ganglia in Embryos and Fetuses

The cranial nerves comprise 12 pairs that have essential sensory and motor functions. Cranial sensory ganglia, derived from neural crest and/or ectodermal placodes, are present on the roots of certain cranial nerves [39,40,41]. Developmental defects in cranial nerves and ganglia are often mentioned as possible causes of death of mutant mice shortly after birth [42,43,44,45]. The simplest and most commonly used method to observe the initial stages of cranial nerve development between E9.5 and E10.5 is to perform whole-mount immunostaining using an antibody recognizing neurofilaments (Figure 3a). Reporter transgenes expressed in the peripheral nervous system can also be used [46]. Cranial nerve development is also interesting to study at later developmental stages, because the bulk of neurogenesis in ganglia normally occurs between E10.5 and E13.5 [47] and several reports have highlighted losses of neuronal populations in ganglia of mutant mice at fetal stages [44,48,49,50,51]. To analyze the late stages of cranial nerve and ganglia development, classical immunohistochemistry or in situ hybridization on histological sections are mostly used, but these methods do not provide 3D information and do not permit one to analyze the nerves in their anatomical environment. Whole-mount immunolabeling methods, combined with 3D imaging of solvent-cleared organs, permits researchers to label tissues in large structures [14,15,16,52,53,54]. However, immunolabeling and the clearing of thick samples is not always feasible as it depends on the availability and robustness of the antibodies and on the nature of the tissues.

We used the Histo3D as an alternative to whole-mount immunohistochemistry to generate 3D reconstructions of cranial nerves and ganglia at different stages of normal mouse development. We manually segmented the trigeminal (V), facioacoustic (VII, VIII), glossopharyngeal (IX) and vagus (X) ganglia and the main cranial nerves at E10.5, E11.5 and E15.5. At E10.5, the trigeminal (Figure 3c,g), facioacoustic, glossopharyngeal and vagus ganglia primordia were readily identified using 2D HREM data. However, compared to the classical immunostaining approach with anti-neurofilament antibodies, the trigeminal, facioacoustic and glossopharyngeal nerves were difficult to detect and only the vagus nerve was clearly discernible. At E11.5, with the progress of neurogenesis, the ganglia were larger and some other nerves could be segmented, including the roots of the trigeminal nerve, connecting to the brainstem, and the three main ophthalmic (V1), maxillary (V2) and mandibular (V3) branches (Figure 3d,h). At E15.5, the trigeminal, facial, glossopharyngeal and vagus ganglia could be readily segmented. The main branches of the trigeminal, facial, glossopharyngeal, vagus and hypoglossal nerves were also conspicuous on the 2D data (Figure 3e,f,i, Appendix A). Vestibulocochlear ganglia and nerves were segmented as a whole (Figure 3e,f,j).

Altogether, these data indicate that for the analysis of neuronal networks, the resolution of HREM images does not reach that obtained by immunohistochemistry. However, the morphological details are sufficient to document the topology and shape of the trunks and ganglia of the cranial nerves from E10.5 to E15.5. Interestingly, 3D reconstructions of HREM data were recently used to characterize the phenotype of *Tubb3* mutants which are dying at birth due to abnormalities in the peripheral nervous system [55].

#### 3.1.3. 3D Reconstructions of Urogenital System Development

We also used Histo3D to investigate changes in the shape and morphology of the urogenital system in the normal mouse at E11.5, E12.5, E13.5 and E14.5 (n = 1 at each developmental stage, all females). The major components of the system, including the gonads, the mesonephros, the metanephros (definitive kidney), the Wolffian (male) ducts and Müllerian (female) ducts, as well as the cloaca, urogenital sinus and ureters, were segmented and reconstructed in 3D.

At E11.5, it was possible to identify and to segment on HREM 2D images the genital ridges, the lumen of the cloaca and the ureteric buds that evaginate from the Wolffian ducts and invade the metanephric mesenchyme on each side of the midline. The reconstruction of the mesonephros was represented in two different ways. Firstly, the mesenchyme adjacent to the gonad and surrounding the Wolffian and Mullerian ducts was segmented as the mesonephric mesenchyme [56]. Secondly, the dense areas within this mesenchyme were segmented as the mesonephric tubules (Figure 4e). At this stage, the genital tubercle was barely visible externally, but it could be observed on 2D HREM sections (Figure 4g) (Appendix A). At E12.5, the cloaca was partitioned, giving rise to the urogenital sinus; the ureters had elongated and branched into the metanephric mesenchyme, but they did not communicate with the urogenital sinus yet; the metanephric mesenchyme had started its anterior migration, and the gonads displayed an elongated cylindrical shape; the mesonephros and the Wolffian could be completely segmented; only the anterior third of the Müllerian ducts had formed at this stage; the genital tubercle was visible in external views of the embryo (Figure 4b) (Appendix A). At E13.5, the urogenital sinus, the metanephros, the ureters (ending at a distance from the urogenital sinus), the gonads, the Wolffian and Müllerian ducts as well as the mesonephros could be segmented (Appendix A). At E14.5, the urogenital sinus was subdivided into the pelvic urethra, connected to the Wolffian ducts, and the urinary bladder, connected to the ureters (Figure 4d,h, Appendix A). Both the Wolffian and Müllerian ducts were still present and complete. The shape of the gonad had changed from cylindrical to ovoid. Mesonephric tubules were still identified (Figure 4f) even though it is generally accepted that the mesonephros has degenerated at E14.5 [57,58]. Interestingly, a similar analytical approach allowed the identification of congenital defects in the urogenital system of retinoic acid receptor-knockout mice [59] (Mark et al., 2021, this issue).

### 3.2. Quantification of Atherosclerotic Plaques in Adult Mouse Aortas

Atherosclerosis contributes to coronary artery disease, stroke, and peripheral vascular diseases and is the leading cause of morbidity and mortality worldwide. Hypercholesterolemic mice such as *ApoE*^−/−^ are one of the most commonly used mouse models of atherosclerosis, as the mice develop complex atherosclerotic plaques on a regular chow diet [60].

Classical methods of quantification, mostly ex vivo, include Sudan IV en face staining preparations for the quantification of lipid-laden plaques in large segments (Figure 5a–c) and serial cross-sections of plaques are often analyzed to measure intimal microscopical lesions [61,62]. These preparations represent the lesions in a 2D manner, with the consequence of underestimating the plaque burden. 3D imaging methods have been developed for atherosclerotic plaque quantification, including high-resolution magnetic resonance imaging (MRI) [63,64,65] and µCT [66].

We used Histo3D to evaluate atherosclerosis development in *ApoE^tm1Unc/tm1Unc^* mice (Figure 5d–f, Appendix A). As an example, we describe the statistical analyses obtained through the treatment of mice with a compound at two different concentrations. *ApoE^tm1Unc/tm1Unc^* mice were randomized into three groups of 12 mice: vehicle, treatment 1 and treatment 2. The aortic roots and branching arteries, namely, the brachiocephalic trunk, left common carotid artery and left subclavian artery, and thoraco-abdominal aortas were used.

Atherosclerotic development was high in the three groups. Statistical analysis of the ‘plaque volume vs. aorta volume ratio’ did not reveal any significant difference between the three groups (Figure 5g). We confirmed that our general method of volume quantification was appropriate by comparing the volumes of aortas only (Figure 5h). No significant difference was detected, showing that the volume measurements of the aortas were consistent between groups. Finally, we also compared the total volumes of atherosclerotic plaques (see Figure 5i). No significant difference was observed, confirming the observation that atherosclerotic development was similar between the three groups.

From a qualitative perspective, the plaques were mainly localized on the wall of the aortic cross and large plaques were very often seen in the walls of the branching cephalic arteries and significantly occluding the lumen, demonstrating the relevance of including these arteries. Some localized plaques were also visualized in the descending aorta. The distribution of the plaques was similar between the three groups.

### 3.3. Morphology and Shape of Ligaments in the Knee Joint of the Adult Mouse

Anatomical exploration of synovial joints of the mouse is difficult to achieve through dissection. Because of the small size and strong relationships of ligaments and capsules with muscular insertions, it is extremely challenging to complete any clean preparation of them without significant alteration. Thus, little is known about mouse synovial joint anatomy. So far, only general considerations about mouse arthrology have been presented in the literature, such as general synovial joint organization and a description of the histological organization of the structures [67,68].

The HREM technology permits the acquisition of high-resolution images upon gross dissections of the joints. With the 3D joint models, one can navigate into and along various planes in space, making it possible to identify and follow ligaments, meniscuses, sesamoid bones, and their relationships with bones. As a proof of concept, we performed 3D reconstructions of the ligaments of the knee in a mouse (Figure 6, Appendix A). Although it is very similar to what is observed in humans (see meniscuses and cruciate ligaments), it was possible to obtain subtle details. For example, we were able to describe the collateral lateral ligament as a bifid ligament, ending for one branch on the head and the hook of the fibula, and for the second branch on the lateral condyle of the tibia. We were also able to observe the presence of menisco-femoral and menisco-tibial ligaments in the posterior region of the intercondylar space. Therefore, this demonstrates how HREM technology is a useful approach to reconstructing the detailed topographic organization of complex anatomical structures, and that it offers unexplored perspectives for the analysis of locomotor system deficiencies.

## 4. Discussion

In this study, we described the development of a new apparatus called “Histo3D” consisting in the combination of the Leica Biosystem Nanocut rotary microtome with a zoom and a camera for HREM experiments. We used this new system to carry out the quantification of different parts of the neuroepithelium and ventricular system at E14.5 and E16.5 stages of mouse fetuses, to visualize cranial nerves and associated sensitive ganglia at E10.5, E11.5 and E15.5 stages and the urogenital system from E11.5 to E14.5 stages. We also showed examples of its utilization to characterize pathophysiological changes at the adult stage, such as for the quantification of atherosclerotic lesions in aortas of hypercholesterolemic mice or the analysis of insertions of ligaments in knee joints of the adult mouse.

We show that the Histo3D system perfectly meets the needs for use as a classical HREM technique, using samples embedded in resin supplemented with acridine orange and eosin, as already shown [20,22,31]. The system can be of great interest to researchers who wish to develop HREM technology in their laboratory, as it offers low-cost and high-quality information using readily available laboratory equipment. For example, Histo3D can be installed on already existing Nanocut microtomes. For our applications, we used a numerical resolution (voxel sizes) between 3.2 × 3.2 × 5 µm^3^ for the smaller samples to 8 × 8 × 7 µm^3^ for the bigger one. These resolutions were obtained after the calibration of the field of view with a micrometer rule for each specific position of the manual zoom (see Table 1). The field of view and slice thickness could be adapted to approach at best the isotropic voxels for every size of the samples and especially for smaller-sized samples. It is noteworthy that with Histo3D, the spatial resolution (measure of detail found in the image) obtained with the bigger samples and by collecting 1000 sections over 3.5 h, permitted the segmentation and reconstruction of structures as thin as cranial nerves (vagus or hypoglossal nerves, for example) or vessels [59] (Mark et al., 2021, this issue), which is comparable to previous reports of cardiovascular or nervous system studies [22,23,27,30,69] using smaller numerical resolution (2 × 2 × 2 µm^3^) and generating more than 2000 to 4000 single sections in 4 to 8 h of sectioning.

As the Histo3D system is very compact and easily upgradable, it will be easier to implement new imaging modalities such as polarizing filters to visualize specific samples such as striated muscle. The Histo3D system is also equipped with multi-wavelength fluorescent filters for the 3D visualization of specifically labeled structures. Although the first publications included examples for visualizing LacZ-stained tissues or whole-mount in situ hybridization, this is still experimental [20,70]. The use of other embedding techniques to visualize fluorescent signals is also opening new fields of applications for Histo3D that are very promising [71]. In this context, the use of the Nanocut rotary microtome with Histo3D will be a powerful technique to process very small samples as it is able to use sections as thin as 1 µm or even less.

The quantification of different parts of the neuroepithelium and ventricular system at specific stages of development may have importance during phenotyping because it can lead to more in-depth analysis in specific brain structures, such as proliferation or apoptosis markers, for example. Our data indicate that, between E14.5 and E16.5, the volume of the structures expanded differently within the neuroepithelium, with the telencephalon growing faster than the other parts of the brain. In contrast, the volume of the ventricles was stable compared to the neuroepithelium, which continued to grow during E14.5 and E16.5. These results are in accordance with previous reports showing that the growth rate of vesicles differs between different species. In human and rats, the forebrain grows faster than in chicks, whereas the mesencephalon grows slower [32]. It may also be interesting to perform such analyses at earlier development stages. Our analyses of E14.5 and E16.5 embryos indicate that there is not much variability in the volumes of the whole neuroepithelium, ventricles and segmented structures. Embryos of the same size or of the same weight were already sufficient to obtain accurate and statistically significant results with a reduced number of animals (three, as shown in this report). The study of the brain at other developmental stages or of other organs may present much more variability and may require more precise staging of the embryos [72,73,74].

Perinatal death is often linked to peripheral nervous system defects that are difficult to describe. The segmentations obtained for cranial nerves and associated sensitive ganglia with HREM data showed that at E10.5, with the resolution that we used, only cranial ganglia were clearly morphologically discernible. One stage later, at E11.5, some nerves connecting the ganglia could be morphologically identified. At E15.5, the main nerves of the trigeminal, facioacoustic, glossopharyngeal and vagus ganglia were observed. Clearly, HREM data could not replace the classical whole-mount anti-neurofilament immunostainings that are generally used to search for cranial nerve abnormalities at early stages of development (from E9.5 to E11.5) but they were capable of being used to identify later abnormalities of cranial nerves and ganglia. As for the earlier stages, cranial ganglia could be easily segmented and quantified, except the vestibulocochlear ganglia and nerves, which were harder to separate. Additionally, the topology and thickness of the main nerves could be determined at stage E15.5. Along this line, recent data from around 500 mutant lines analyzed with HREM at E14.5 during the Deciphering the Mechanisms of Developmental Disorders (DMDD) program showed that hypoglossal nerve (twelve cranial nerve) abnormalities could serve as a biomarker of CNS defects [27]. The optimization of a pipeline dedicated to models exhibiting perinatal lethality, combining topological analyses of cranial nerves and ganglia using anti-neurofilament staining at early stages and further explorations at later stages using HREM, will be an asset. If the progression of cranial ganglia needed to be explored, it would be possible to initially quantify their volumes at early and late stages of development by means of HREM and according to the results, to perform proliferation and apoptosis-related studies using classical immunohistochemistry analyses.

Histo3D imaging was very helpful in studying the development and remodeling of a complex system such as the urogenital system during embryonic development. Especially, focusing on duct branching (the ureter) is interesting as it was shown that the development of the kidney is initiated when the ureteric bud (UB) branches from the Wolffian duct and invades the overlying metanephric mesenchyme [75]. Furthermore, abnormalities can be depicted in mesonephric (Wolffian) and paramesonephric (Müllerian) duct formation or any other anatomical structure of the system. Additionally, hypoplasia of the gonads or the metanephros could be easily confirmed through volume quantification comparisons of the wild-type and mutant cohorts.

The quantifications of atherosclerotic plaques in the aorta showed that HREM technology is of major interest compared to the classical en face staining because volumetric assessment could be performed. Additionally, HREM data allowed the exploration of the histological organization of the atherosclerotic plaques in situ, compared to other 3D imaging methods such as MRI or µCT [64,65,66]. Indeed, with HREM data, the percentage of stenosis could be evaluated by determining the ratio between arteries’ lumen surface and the plaque surface. Analysis of the free borders of the plaques provides information on the probability of plaque rupture along the whole aorta and branching arteries. Our HREM analyses were performed on the entire aortas, including the aortic roots, aortic arch, branching arteries and thoracoabdominal part. Statistically, the majority and larger-volume plaques were located on the aortic arch and connecting arteries, whereas fewer plaques were observed on the thoracoabdominal aorta. This confirms the importance of analyzing the whole aortas and branching arteries [76]. The statistical analyses illustrated quite high variability in plaque volumes between individuals from the same group, which is intrinsic to the development of atherosclerotic plaques in the *ApoE*^−/−^ mouse model and is not inherent to the technology or quantification method used. Nevertheless, reliable conclusions can be obtained using higher numbers of animals per group. In some cases, immunohistochemical methods may be necessary to quantify the key cellular components of the plaque and other aortas may have to be added to perform those analyses. Another option is to take different parts of the aortas for different purposes, as already described [76].

Whole-body morphological assessment of a GEM is typically accomplished through a pathological review in which a gross necropsy examination and a systematic histological review of major tissues and organs are performed [77,78]. Additional 3D techniques can be of great value in understanding complex morphological processes. Here, we created 3D reconstructions of the ligaments of the knee in adult mice using HREM technology. Although their organization is very similar to that of humans, it was possible to observe subtle details, offering new possibilities for phenotyping purposes in animal models of locomotor system pathologies.

## 5. Conclusions

We used our new Histo3D HREM-based imaging system in a large variety of biological contexts, to demonstrate how this methodology can provide further characterizations of normal and abnormal anatomy in the mouse during prenatal stages and adult life. 3D reconstructions of structures obtained after their segmentation based on the original stacks of images present two main advantages. First, they can highlight shapes and microanatomical structures that cannot be described by dissection, with conventional 2D techniques or with lower-resolution 3D imaging techniques. Secondly, they permit quantitative analyses during normal mouse embryonic development and the analysis of mutant phenotypes. New phenotyping projects envisaged with Histo3D technology should be considered in terms of feasibility concerning the size and type of samples compared to other techniques. Although we have demonstrated the interesting features of our Histo3D-HREM technology, it should still be considered an emerging technique. Refinements and improvements are still needed, especially for visualizing gene expression patterns (chromogenic or fluorescent markers) in various types of samples.

## 6. Construction of Histo3D

The Histo3D was constructed using the optomeca platform of IGBMC. Interested laboratories can directly contact Olivia Wendling for scientific and technical aspects or Didier Hentsch for system requests.

## Figures and Tables

**Figure 1 biomedicines-09-00767-f001:**
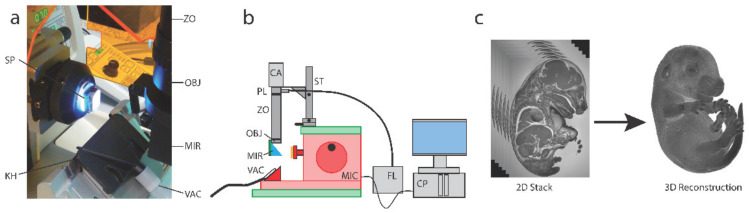
The microtome-integrated Histo3D optical system. (**a**) Detailed view of the block positioning for image capture. The block is positioned above the knife holder (KH). The excitation light passes through the zoom (ZO), the objective length (OBJ) and the mirror (MIR) to illuminate the sample (SP). The produced section is collected by the vacuum cleaner for each cutting-imaging loop. (**b**) The Histo3D apparatus is composed of the Leica Biosystem Nanocut Microtome (MIC) for the block sectioning step. The optical system is installed on a custom XYZ stage (ST) on the top of the microtome. The optical path is based on a custom fluorescence Ploem illuminator (PL) installed on a zoom (ZO) equipped with an objective lens (OBJ). To reduce the system footprint, the optical axis contains a mirror (MIR) in front of the objective lens for both sample highlighting and image collection. The block-face images are collected on the camera (CA). The fluorescence light is provided by a bright solid-state white light source (FL). The image acquisition and sectioning loop is computer-controlled using the Imagxcell Software Suite (CP), allowing the Histo3D system to perform block-face imaging of the sample with a fully automated procedure. For each cutting iteration, the produced section is collected by the vaccum system, equipped with HEPA filters. The Histo3D system does not modify the microtome and thus can be installed on an already existing Nanocut microtome. (**c**) Generation of 2D raw data images and 3D reconstruction of a whole mouse embryo at E15.5.

**Figure 2 biomedicines-09-00767-f002:**
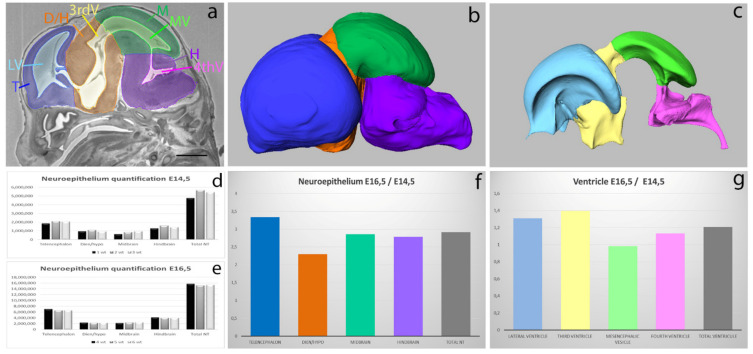
Quantification of neuroepithelium tissue and ventricular system in E14.5 and E16.5 mouse fetuses. Manual segmentation of different structures of the neuroepithelium and ventricles on HREM 2D raw data (**a**). 3D reconstructions of the neuroepithelium (**b**) and ventricles (**c**) at stage E14.5. Quantification of neuroepithelium tissue at E14.5 (**d**) and E16.5 (**e**). The *x*-axis represents the structures in the 3 animals and the *y*-axis represents the voxel size (represented in pixel numbers). Representation of the ratios E16.5/E14.5 in the neuroepithelium (**f**) and ventricle (**g**) volumes. The *x*-axis represents the structures and the *y*-axis represents the volume ratios. T: telencephalon, D/H: diencephalon and hypothalamus, M: midbrain, H: hindbrain, LV: lateral ventricles, 3rd V: Third ventricle, MV: mesencephalic ventricle, 4th V: fourth ventricle. Scale bar: 1 mm.

**Figure 3 biomedicines-09-00767-f003:**
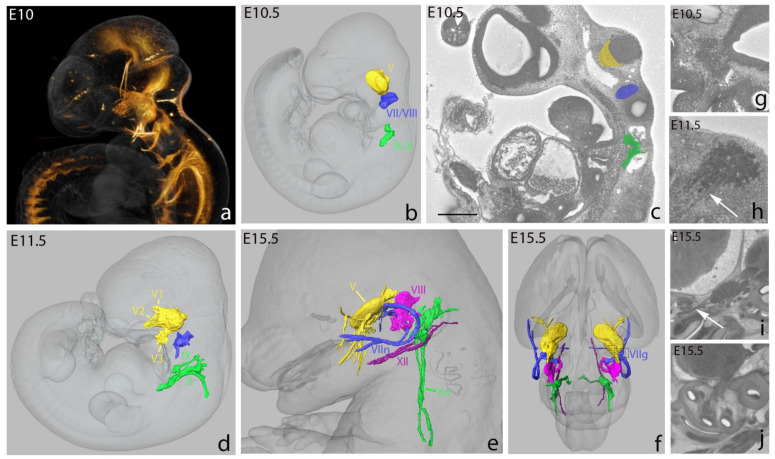
3D representation of cranial nerves and ganglia at E10, E10.5, E11.5 and E15.5. Whole-mount immunostaining of an E10.0 embryo with an antineurofilament antibody (**a**). 3D reconstruction of the ganglia at E10.5 (**b**), the ganglia and cranial nerves at E11.5 (**d**) and E15.5 (**e**,**f**). The whole embryo, head and brain are represented in shades of gray. Segmentation of ganglia at E10.5 on HREM 2D data (**c**). Visualization of trigeminal ganglion (V) at E10.5 (**g**), trigeminal ganglion and nerves (white arrow) at E11.5 (**h**) and E15.5 (**i**). Note the absence of visible nerve fibers at E10.5 compared to E11.5 (red arrow). Visualization of vestibulocochlear ganglia and nerves (VIII) on 2D data (**j**): note the difficulty of segmenting ganglia and nerves at these developmental stages. V: trigeminal ganglion, V1, V2, V3: ophthalmic (V1), maxillary (V2) and mandibular (V3) branches of the trigeminal nerve, VII/VIII: facioacoustic ganglion complex, VII: facial ganglion, IX/X: glossopharyngeal and vagus ganglion complex, IX: glossopharyngeal ganglion, X: vagus ganglion, Xn: vagus nerve, XII: hypoglossal nerve. Scale bar in (**c**): 350 μm, (**g**): 350 μm, (**h**): 200 μm, (**i**): 900 μm, (**j**): 600 μm.

**Figure 4 biomedicines-09-00767-f004:**
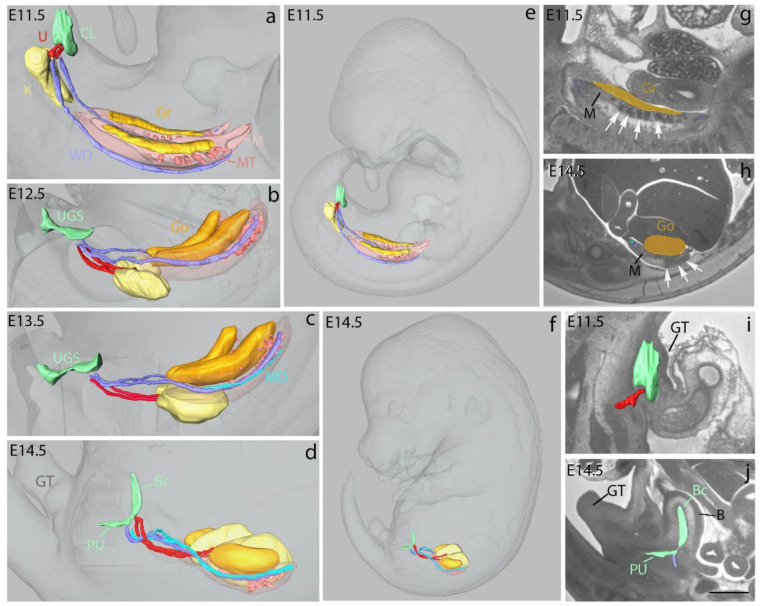
3D representation of the developing urogenital system of mouse embryos. 3D representation of the urogenital system (in colors) and the external surface of the embryos (in shades of gray) at E11.5 (**a**), E12.5 (**b**), E13.5 (**c**) and E14.5 (**d**). Hindlimbs were removed to better visualize the genital tubercle (GT). 3D representation of urogenital system in a whole embryo at E11.5 (**e**) and E14.5 (**f**) to show the changes in topology and shapes during development. Segmentation of the genital ridge (E11.5), gonad (E14.5) (orange), mesonephric mesenchyme (M) and mesonephric tubules (MT) (white arrows) at E11.5 (**g**) and E14.5 (**h**). 3D representation of the urogenital sinus (UGS) and formation of bladder and ureters on sagittal HREM sections at E11.5 (**i**) and E14.5 (**j**). The genital tubercle (GT) is beginning to form at E11.5 and well developed at E14.5. The urogenital sinus comprises the pelvic urethra (PU) and urinary bladder (B) at E14.5 (**d**,**j**). K: metanephros (definitive kidney), Go: gonad, Gr: genital ridge, WD: Wolffian duct, MD: Müllerian duct, CL: lumen of cloaca, U: ureteric bud (E11.5) or ureters, Bc: urinary bladder cavity. Scale bar in (**j**): 150 μm (**i**,**g**), 600 μm (**j**), 850 μm (**h**).

**Figure 5 biomedicines-09-00767-f005:**
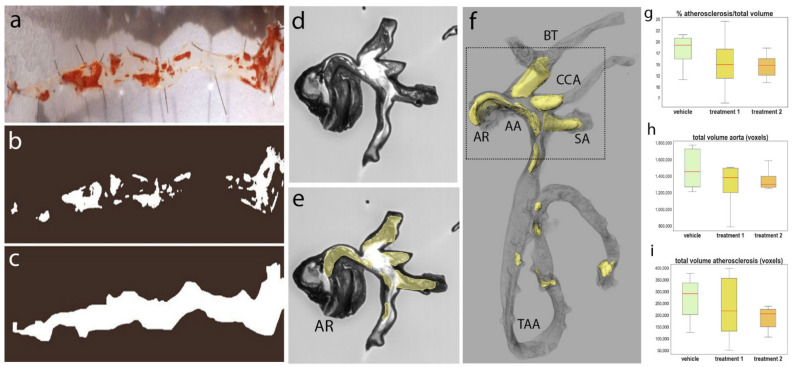
Quantification of atherosclerotic plaques in adult mouse aortas. Representative en face micrograph of an aorta with lipid-laden plaques stained with Sudan IV (in red color) (**a**). The plaque surface (**b**) and whole aorta surface (**c**) are calculated using thresholding and mask generation in image J software. Visualization of atherosclerotic plaques on the original stack of images generated by HREM (**d**). Sagittal section plane has been chosen for illustration. Manual segmentation of the atherosclerotic plaques using Avizo software (yellow) (**e**). 3D reconstruction of the atherosclerotic plaques (yellow) and aorta (gray) (**f**). Quantifications of plaque volume vs. aorta volume ratio (**g**), total aorta volume (**h**) and total volume of atherosclerotic plaques (**i**). The *x*-axis represents the groups of compounds, the *y*-axis represents the plaque/aorta volume ratios, the total aorta volume and total plaque volume, respectively. Red bars represent the median of the distribution and vertical bars represent the standard deviation. AR: aortic root, AA: aortic arch, TAA: thoracoabdominal aorta, BT: brachiocephalic trunk, CCA: left common carotid artery, SA: left subclavian artery.

**Figure 6 biomedicines-09-00767-f006:**
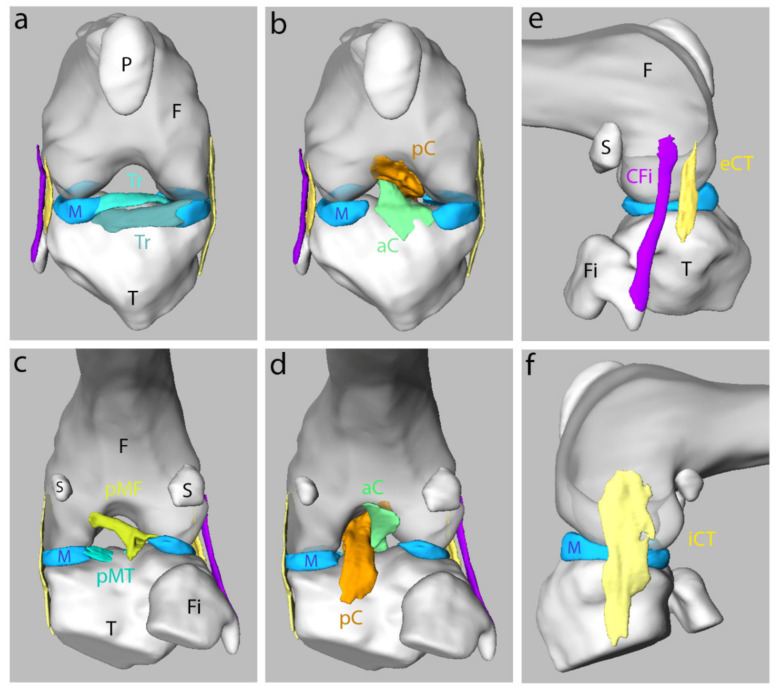
3D representation of the ligaments and bones in an adult mouse knee joint. Anterior (**a**,**b**), posterior (**c**,**d**) and lateral views (**e**,**f**). The meniscus (M) is represented in light blue. F: femur, T: tibia, Fi: fibula, P: patella, S: sesamoid bones, pMF: posterior menisco-femoral ligament, pMT: posterior menisco-tibial ligament, iCT and eCT: internal and external tibial collateral, aC and pC: anterior and posterior cruciate ligament, Tr: transverse ligament, CFi: fibular collateral ligament.

**Table 1 biomedicines-09-00767-t001:** Histo3D pixel size and field of view for specific Z16 zoom positions.

Zoom Position	Pixel Size (µm)	*x*-Axis FOV (mm)	*y*-Axis FOV (mm)	FOV (mm^2^)
0.6	11.7	24	24	574.1
0.8	8	16.4	16.4	268.4
1	6.5	13.3	13.3	177.2
1.2	5.2	10.6	10.6	113.4
1.6	4	8.2	8.2	67.1
2	3.2	6.5	6.5	42.9
2.5	2.6	5.3	5.3	28.3
3.2	2	4.1	4.1	16.8
4	1.6	3.3	3.3	10.7
5	1.3	2.7	2.7	7.1
6.3	1.1	2.2	2.2	5.1
8	0.8	1.6	1.6	2.7
9.2	0.7	1.4	1.4	2.1

## Data Availability

Most data presented in this study are available on request from the corresponding author. Some restrictions apply to the availability of the data from the “Qualitative and quantitative morphological assessment in mouse developmental pathology studies” and the “Quantification of atherosclerotic plaques in adult mouse aortas” but these can be made available upon request respectively from L.L.C. or T.R. with the permission respectively of “CNRS” or “Merck and Co”.

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
