# Peer review of "High Resolution Episcopic Microscopy for Qualitative and Quantitative Data in Phenotyping Altered Embryos and Adult Mice Using the New “Histo3D” System"

_biomedicines, 2021, doi:10.3390/biomedicines9070767_

Round 1

Reviewer 1 Report

Authors report the setup of a “High Resolution Episcopic Microscopy” (HREM) by integration of the rotary microtome with optics and camera for high resolution of serial sectioning images and the reconstruct as 3D reconstructed images for qualitative and quantitative measurement of various structural morphological changes for development and disease models. Using normal and genetically altered embryos and adult organisms in animal models during development. Exemplar quantification of neural tissue and ventricular system volumes of normal brains, cranial and peripheric nerves, urogenital system, atherosclerotic plaques, knee ligaments in adult mice, are demonstrated clearly and logically with control and mutated animal models.

The paper is presented in clear and logical structure with good quality of figures and supported data. The proposed novel system is a good helping tool for studying the biological development. There are some minor issues for possible improvement. 1. A flow chart of image processing procedure using Image J for easier comprehension. 2. A sectioning image in the Figure 1 from microtome stage will be easier to understand the source of 2D images for 3D reconstruction. Overall, I would recommend to publish this manuscript in current format with supportive results or with minor amendment with suggestions. 

Author Response

We thank the reviewer for his positive comments and we tried to answer the two points raised for improving the manuscript: 

  1. A flow chart of image processing procedure using Image J for easier comprehension.

The “3D  Mode” of the microtome of the Histo3D system permits to obtain perfect realignements of the sections without post processing of the images for 3D reconstruction. Hence, the use of Image J software to process the raw data images is limited to the cropping of  the sample, the adjustment of the brightness and contrast and an eventual scaling of the big stacks. This is mentioned in the Material and method section 2.2 (in grey). We added one sentence in the text (highligted yellow) for further comprehension. 

lane 146 : "2-D raw data were imported into the Avizo visualization software (Version 9.4.0, Thermofisher Scientific, France) for volumetric visualization, segmentation, and quantitative analysis. Before importation, the images were cropped, eventually scaled and the contrast optimized using Image J. No realignement of images was necessary due to the precise positioning of the bloc after each section. "

As no real procedure is in place with Image J software , we propose that a flow chart is not necessary. 

  1. A sectioning image in the Figure 1 from microtome stage will be easier to understand the source of 2D images for 3D reconstruction

Section C was added on Figure 1 to illustrate 2D raw data images and 3D reconstruction on an embryo at E15.5 . The text has been added in the legend and in the results (in yellow).

lane 140: (c) Generation of 2D raw data images and 3D reconstruction of a whole mouse embryo at E15.5.

Reviewer 2 Report

This is an interesting work and the author used new Histo3D HREM-based imaging system to demonstrate how this methodology can provide further characterisation of normal and abnormal anatomy in the mouse during prental stages and adult life. I think this work would be benificial for the community. I recommend to be accpeted. 

Author Response

We thank the reviewer for it supporting comment